# Arabic Language Opinion Mining Based on Long Short-Term Memory (LSTM)

Arief Setyanto [1,*], Arif Laksito [2,*], Fawaz Alarfaj [3,*], Mohammed Alreshoodi [4], Kusrini [1], Irwan Oyong [2], Mardhiya Hayaty [2], Abdullah Alomair [3], Naif Almusallam [3] and Lilis Kurniasari [5]

[1] Magister of Informatics Engineering, Universitas Amikom Yogyakarta, Yogyakarta 55281, Indonesia; kusrini@amikom.ac.id

[2] Faculty of Computer Science, Universitas Amikom Yogyakarta, Yogyakarta 55281, Indonesia; oyong@amikom.ac.id (I.O.); mardhiya_hayati@amikom.ac.id (M.H.)

[3] Department of Computer and Information Sciences, Imam Mohammad Ibn Saud Islamic University, Riyadh 11564, Saudi Arabia; amalomair@imamu.edu.sa (A.A.); nyalmusallam@imamu.edu.sa (N.A.)

[4] Department of Natural Applied Science, Applied College, Qassim University, Buraydah 52571, Saudi Arabia; mo.alreshoodi@qu.edu.sa

[5] Departemen of Electrical Engineering, Universitas Nahdlatul Ulama Yogyakarta, Yogyakarta 55162, Indonesia; lilis@unu-jogja.ac.id

\* Correspondence: arief_s@amikom.ac.id (A.S.); arif.laksito@amikom.ac.id (A.L.); fkarfaj@imamu.edu.sa (F.A.); Tel.: +62-81316024569 (A.S.); +62-8562926468 (A.L.); +966-503901500 (F.A.)

**Abstract:** Arabic is one of the official languages recognized by the United Nations (UN) and is widely used in the middle east, and parts of Asia, Africa, and other countries. Social media activity currently dominates the textual communication on the Internet and potentially represents people's views about specific issues. Opinion mining is an important task for understanding public opinion polarity towards an issue. Understanding public opinion leads to better decisions in many fields, such as public services and business. Language background plays a vital role in understanding opinion polarity. Variation is not only due to the vocabulary but also cultural background. The sentence is a time series signal; therefore, sequence gives a significant correlation to the meaning of the text. A recurrent neural network (RNN) is a variant of deep learning where the sequence is considered. Long short-term memory (LSTM) is an implementation of RNN with a particular gate to keep or ignore specific word signals during a sequence of inputs. Text is unstructured data, and it cannot be processed further by a machine unless an algorithm transforms the representation into a readable machine learning format as a vector of numerical values. Transformation algorithms range from the Term Frequency–Inverse Document Frequency (TF-IDF) transform to advanced word embedding. Word embedding methods include GloVe, word2vec, BERT, and fastText. This research experimented with those algorithms to perform vector transformation of the Arabic text dataset. This study implements and compares the GloVe and fastText word embedding algorithms and long short-term memory (LSTM) implemented in single-, double-, and triple-layer architectures. Finally, this research compares their accuracy for opinion mining on an Arabic dataset. It evaluates the proposed algorithm with the ASAD dataset of 55,000 annotated tweets in three classes. The dataset was augmented to achieve equal proportions of positive, negative, and neutral classes. According to the evaluation results, the triple-layer LSTM with fastText word embedding achieved the best testing accuracy, at 90.9%, surpassing all other experimental scenarios.

**Keywords:** sentiment analysis; opinion mining; Neural Network; LSTM; Arabic

## 1. Introduction

Natural language processing (NLP) has become a popular research topic in the last two decades. Its popularity has attracted researchers to examine data, web and text mining, as well as information retrieval [1]. Opinion mining is one attractive topic, among other

topics such as machine translation, customer support boss, text summarization, and speech recognition. Opinion mining is essential in current online interactions, such as e-commerce, e-government, and many online services, for understanding public opinion on specific issues [2]. Governments worldwide increasingly adopt social media to engage their citizens and understand citizen opinion with respect to public policy [3].

Opinion mining can mainly be divided into lexicon-based and supervised machine learning approaches. Lexicon-based approaches rely on the sentiment word dictionary. To carry out lexicon-based sentiment analysis, lexicon dictionaries are available for public use, such as wordnet in English, [4] in Bahasa, and [5–8] in Arabic. On the other hand, the supervised approach needs manually labeled corpora to train the classifier. Researchers are working on lexicon-based opinion mining in many languages, such as [9] for the Indonesian language, [6–10] for Arabic, and [11] for Urdu. The supervised approach has attracted researchers in detecting sentiment levels using SVM, ANN, Naïve Bayes, Decision Tree, Random Forest, and numerous exercises with many different languages and datasets. Currently, researchers are moving forward in implementing deep learning for natural language processing in general, specifically with respect to sentiment polarity classification.

The computing power available in current hardware and cloud services allows researchers to solve problems using heavy computing algorithms. This capability has enabled a massive development in deep learning as an extension of the neural network concept. Deep learning has made a successful breakthrough in many recognition tasks for image, video, and text. In particular, language signals have specific properties regarding time sequence. The early research on textual recognition assumed that the sequence had no significant effect. This assumption led to a lower recognition rate for early sentiment classification using time-series-unaware algorithms such as Artificial Neural Network, Support Vector Machine, and Naïve Bayes. Long Short-Term Memory (LSTM) extends an artificial neural network with a connection between an earlier neuron and the next neuron. Because the neuron in LSTM does not only contain a single operation, it is called an LSTM-cell instead of a neuron. With LSTM architectures, if the sentence in a corpus consists of five words, the first word will come into the first unit, while subsequent words will go to the second through fifth units. To maintain the contribution of the first LSTM unit in the computation of the second unit, the output of the first unit needs to be connected to the second unit. This connection is maintained between subsequent calls. LSTM ensures that word signals in a sentence affect each subsequent word. Therefore, LSTM is suitable for the computation of time series problems, textual and time series prediction analysis, sound, and voice computation. With respect to problem construction, this paper discusses the implementation of LSTM for sentiment analysis.

An extensive dataset is needed in a deep learning environment to enable the necessary training to obtain the pattern of the problem. Deep learning implementation in image and video processing has benefited from the huge datasets available to the public. For instance, the ImageNet (https://www.image-net.org/index.php, (accessed on 1 February 2022)) dataset for visual recognition and the YouTube 8M (https://research.google.com/youtube8m/, (accessed on 1 February 2022)) dataset for video understanding have been vital for deep learning algorithms to make perform accurate recognition. Image and video recognition are generally applied for many purposes, regardless of geographical and cultural boundaries. Text datasets, however, cannot be treated generally. For example, a textual dataset for Arabic cannot be used to train English sentiment analysis or vice versa. Therefore, natural language processing research requires many adjustments due to cultural boundaries.

The textual signal naturally consists of non-numeric and unstructured data. Processing unstructured data is a challenge for researchers in NLP, and therefore many researchers have proposed formulas for transforming the unstructured data into a more structured format. A bag of words has been proposed to transform textual data into numerical data by exploiting the frequency of each word in a sentence, resulting in the concepts of Term Frequency and Inverse Document Frequency proposed in [12]. Word embedding methods have been proposed to obtain more redundant data in order to represent words in

consideration of their relationship to other words in many contexts. There are a number of word embedding techniques, such as Word2vec [13], fastText [14], and GloVe [15]. Word embedding requires a massive corpus in order to train the algorithms to calculate the vector values of specific words.

Textual signals may come from a number of unofficial sources, such as social media, e-commerce comments, book reviews, and websites. This condition leads to data problems such as duplicated, unnecessary, meaningless, non-mixing, and stop words. Normalizing data before further processing plays a significant role in achieving high classification accuracy. In many languages, such as in English or Indonesian, the meaning of a word rarely changes with the addition of prefixes or suffixes. Therefore, removing prefixes and suffixes can help reduce the number of specific words (terms) in a corpus, thus reducing the computing complexity. However, in Arabic, words may change their meaning entirely with the addition of a prefix or suffix; therefore, stemming might lead to misinterpretation of the sentence.

The Arabic language is one of the most widely spoken languages today. It is an official language of the United Nations organization, estimated to be used daily by more than 400 million people. Its usage on the web has shown vigorous growth in recent years, where it ranked as the fourth most used language in cyberspace [16].

The Arabic language can be found in three forms: Classical Arabic (CS), which adheres to strict grammatical and morphological rules and is usually used in literary texts. Second, Modern Standard Arabic (MSA) is commonly used in correspondence and formal speech. Lastly, Dialectical Arabic (DA) refers to oral utterances spoken in daily communication [17]. Typically, microblogging content will be in MSA or DA or a variant of the two where the Arabic words are written using Latin letters, numbers, and punctuation [18].

This research observes varied embedding methods and Long Short-Term Memory (LSTM) with single, double and triple layers to recognize sentiment polarity. The authors adopt the Arabic Sentiment Analysis (ASAD) dataset from [19] to evaluate the proposed frameworks.

## 2. Literature Review

Several studies have been conducted on sentiment analysis using various algorithms in different languages such as Indonesian [9], Urdu [13], Russian [20], and Arabic [21]. However, most work has been carried out in English, such as [22–24]. The study of natural language processing is interesting for two reasons, firstly due to the algorithm's variations, and secondly due to language variety. A successful framework for a particular language may need adjustment when it comes to implementation in a different language. Sometimes, even dialect can heavily affect the likely result. In Arabic, for example, the researchers in [25] divide the variety of languages into three groups: Classical Arabic (CA), Modern Standard Arabic (MSA) and Dialectical Arabic (DA).

Sentiment analysis is a classification task of input text into several classes. The number of classes varies from binary (positive, negative), three classes (binary with neutral) and complex systems of five classes ranging from very positive to very negative. Arabic is one of the most popular languages in the world, with 330 million native speakers, and is one of the official languages of the United Nations (UN) [26]. There are some public datasets available in the Arabic language of various classes. Some public datasets consist of positive and negative classes such as the Large-Scale Arabic Book Review [27] and Ar-Twitter, proposed by [28]. The rest of the available dataset consists of four more classes, such as [29], which proposed four classes, and ArsenTb, which employs five classes [10,30,31].

The sentiment analysis is mainly divided into lexicon-based and supervised machine learning approaches. Lexicon-based approaches rely on the sentiment word dictionary. Some researchers have claimed promising results using lexicon-based approaches, such as [32,33]. Supervised approaches rely on machine learning algorithms and are based on a labeled corpus; therefore, they are also referred to as corpus approaches. The machine

learning algorithms are basically trained by the labeled corpus to build a model. There are many researchers who have employed a variety of languages and algorithms.

The lexicon approach was quite popular at the beginning of sentiment analysis study. It is powerful and can be performed using simpler algorithms. However, it is mainly based on the availability of lexicon dictionaries. A pre-defined dictionary provides a set of words in each sentiment polarity. A document is classified into a particular polarity on the basis of the word frequencies of each polarity side from the dictionary. Ref. [34] compared six sentiment opinion lexicons, and proposed a new general-purpose sentiment lexicon that they claimed was able to achieve 69% accuracy when determining the sentiment of news headline. In sentiment analysis in the Arabic language, ref. [32] proposed a sentiment lexicon with a size of 16,800 words. Their experiment claimed that the integrated lexicon achieved better results, at 74% accuracy, than manual and dictionary-based sentiment analysis. Later on, ref. [33] proposed a sentiment lexicon consisting of 120,000 Arabic terms. They claimed an even better accuracy of 86.89%. The lexicon approach generally achieves better performance when the sentiment lexicon is complete. However, as a language is not static knowledge, this approach needs to be continuously updating the data all the time. Therefore, researchers have tried to find a better way of overcoming the language dynamics. Corpus-based methods rely on real-time data such as reviews, social media, and the web. By annotating the real-time data, data researchers train the machine to recognize the patterns of each sentiment polarity.

The supervised approach relies on machine learning algorithms such as support vector machines, decision trees, logistic regression, and many more. Ref. [35] compared the Support Vector Machine (SVM) and Artificial Neural Network (ANN) for detecting sentiment on 3000 sentences from the Movie Review dataset and 3000 customer reviews on Amazon for particular products. According to their experiments, ANN performed significantly better than SVM on sentiment classification. Ref. [24] carried out a study on the scalability of the Naïve Bayes classifier for big data, and found the Naïve Bayes classifier to be capable of analyzing millions of movie reviews with 82% accuracy using a vast dataset. Although Arabic is a less-researched area with respect to sentiment mining, there have been some reports. Ref. [36] compared the performance of five classifiers: Support Vector Machine, Random Foes, Gaussian Naïve Bayes, Logistic Regression, and Stochastic Gradient Descent. They also compared the Skip-Gram model and Continuous Bag of Words (CBOW) by fastText as vectorization methods. They found that fastText Skip-Gram performed better for all classifiers. Ref. [37] examined Hierarchical Classifier, SVM, Decision Tree (DT), Naïve Bayes, K-Nearest Neighbor (KNN) on an extensive book review dataset. They found that their Hierarchical Classifier achieved 57.8% accuracy. Ref. [38] compared Convolutional Neural network (CNN), Naïve Bayes, Logistic Regression and Support Vector Machine (SVM) for sentiment classification on a health dataset. According to their experiments, SVM gave the best results. However, there was no common agreement among researchers regarding high-performance algorithms dealing with sentiment analysis on the basis of those experiments. Some algorithms may work very well under certain conditions, but fail on the problem set.

The aforementioned supervised classifier fails to capture word order sequences in the text. Most machine learning algorithms cannot distinguish differences in word order, such as between "the dog kills a mouse" and "the mouse kills a dog". A machine learning architecture is needed that is aware of time sequences. Recurrent Neural Network (RNN) has connections between previous, current and future signals, and therefore it is fit to represent time-series signals, including text. Ref. [39] proposed an implementation of RNN called LSTM (Long Short-Term Memory) in order to retain the long-term memory effect.

Some researchers have implemented RNN and LSTM in many languages, including Arabic, as shown in Table 1. Ref. [9] proposed an Indonesian sentiment corpus and classification engine using word2vec and LSTM. In Ref. [40], proposed Hybrid CNN-LSTM Model outperformed traditional deep learning and machine learning techniques in precision, recall, f-measure, and accuracy. Hybrid CNN-LSTM was tested on the IMDB

sentiment dataset and achieved an improved accuracy of 91%. Ref. [41] explored the performance of a deep learning framework for an Arabic corpus of around 40,000 tweets using Word2vec and several architectures. According to their experiments, LSTM with data augmentation to balance the dataset overperformed compared to LSTM without data augmentation, and the CNN and RCNN models. Ref. [42] implemented CNN for feature extraction and LSTM to capture long-term word dependency. They achieved 64% accuracy on three-class sentiment prediction. Ref. [41] focused on a health services sentiment dataset. They used the English dataset translated into Arabic, carried out the classification using RCNN, and achieved 94% prediction accuracy. Ref. [30] implemented LSTM on a small corpus with five classes in two Arabic dialects: Emirati and Egyptian. They achieved accuracies of 70% on Egyptian dialects and 63.7% on Emirati dialects. Ref. [43] performed an exercise using multiple datasets, word embedding methods, various classic machine learning methods, and a deep learning framework. They relied on fastText word embedding for deep learning using CNN, LSTM, and bidirectional LSTM. Ref. [21] explored the Recursive Neural Tensor Networks (RNTN) model using the word2vec word embedding method. Ref. [44] explored CNN and two-layer LSTM. Their best achievement was recorded at an accuracy of 90.75% on the fastText Skip-Gram CNN-LSTM framework. Among those deep learning approaches mentioned above, there are many aspects, such as pre-processing, word embedding methods, deep learning architectures, and dataset composition. There is no silver bullet approach that solves all problems, and therefore deeper exploration using a variety of datasets, pre-processing methods, architectures and word embedding methods remains chellenging. Table 1 lists several studies on Arabic sentiment analysis, the algorithms used, and their achieved accuracy.

**Table 1.** Arabic Sentiment Classification Literature review.

| Research Papers | Dataset | Composition | Language | Algorithm | Accuracy |
|---|---|---|---|---|---|
| [41] | Corpus on Arabic Egyptian tweets | 40,000:20,000 Positive, 20,000 Negative | Arabic (Modern Standard Arabic) and Egyptian Dialect | LSTM + Augmentation | 88.05 |
| | | | | LSTM | 81.3 |
| | | | | CNN | 75.72 |
| | | | | RCNN | 78.76 |
| [44] | LABR (Large Scale Arabic Book Review) | 63,257:5 classes 1:2939, 2:5285, 3:12,201, 4:19,054, 5: 25,778 | Arabic | FastText Skip gram—CNN-LSTM | 90.75 |
| | | | | LSTM 1 layer | 87.98 |
| | | | | LSTM 2 Layers | 90.75 |
| [36] | LABR (Large Scale Arabic Book Review) | 63,257:5 classes 1:2939, 2:5285, 3:12,201, 4:19,054, 5:25,778 | Arabic | Logistic Regression | 84.97 |
| [42] | Arabic SemEval-2016 dataset for the Hotels domain | 24,028 | Arabic | CNN | 82.7 |
| | | | | LSTM | 82.6 |
| [43] | Arabic SemEval-2016 dataset for the Hotels domain | 24,028 | Arabic | Bi-LSTM | 87.31 |

## 3. Methodology

### 3.1. Frameworks

In this research, we propose to employ a framework using pre-trained word embedding and training net LSTM. Figure 1 shows the framework for Arabic text classification. Moreover, this research aims to understand the effect of word embedding algorithms. fastText [14] and GloVe [15] are compared in both frameworks. Glove [15] and FasText [14] are advanced developments of Word2vec [13]. Glove was developed to capture global context; however, most embedding techniques, including Glove, to fail to capture incomplete words or subwords with slightly different meanings. According to [15], fastText was

developed to capture the internal structure of the words in morphologicaly rich languages, including Arabic. FastText, on the other hand, was trained not only using words, but also with subwords using n-grams. For example, the word "Eating" would be considered as "eating", "eat" and "ing" during training. This process leads to higher computing load during the training process; however, it successfully deals with incomplete words and chanes in word form. The Arabic language has some morphological operations that slightly affect meaning, such as by changing the word "kataba" (write), into "kitaabun" (book), and "maktabun" (place to write). Therefore, this research observes the impact of implementing a morphologically aware embedding technique (fastText) compared to Glove.

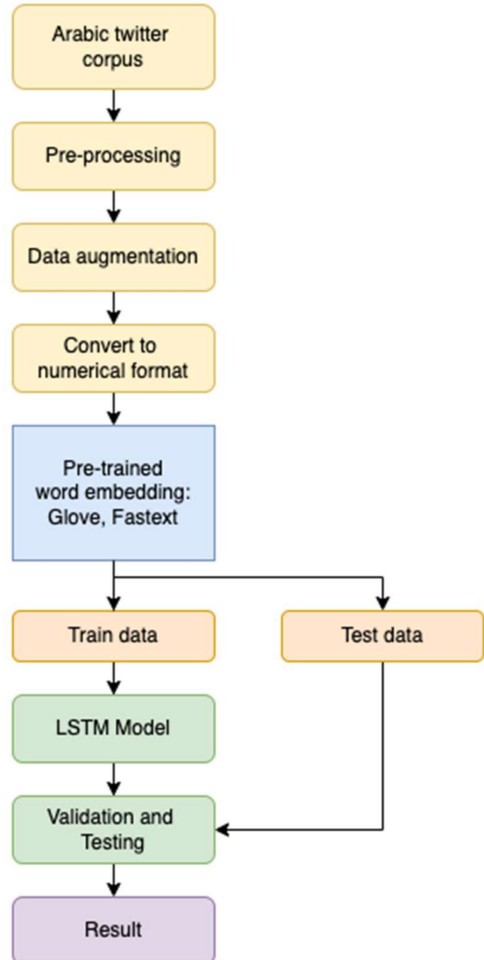

**Figure 1.** Proposed framework for Arabic text classification.

*3.2. Experimental Scenarios*

This research aims to observe the impact of word embeeding methods and the LSTM architectures. To observe the impact of pre-processing and LSTM architectures, a step-by-step task is defined and divided into six scenarios. The effect of the embedding methods and the impact of the LSTM architectures can be observed using the scenario in Table 2.

Training and evaluation were carried out using early stopping and k-fold cross-validation. Early stopping was implemented with a threshold set at 18 to obtain the model. The training was stopped when the accuracy did not improve after 18 epochs. The final evaluation was performed by means of 5-fold cross-validation. We took the average of the accuracy and the standard deviation as indications of the quality of the results of the classification task.

**Table 2.** Research scenarios.

| Word Embedding | LSTM Architectures |
|:---:|:---:|
| | Single-Layer LSTM |
| GloVe | Double-Layer LSTM |
| | Triple-Layer LSTM |
| | Single-Layer LSTM |
| fastText | Double-Layer LSTM |
| | Triple-Layer LSTM |

*3.3. Word Embedding*

3.3.1. GloVe

Word embedding aims to transform textual information into a vector of real value. Semantic/language vector space models of language represent each word with a real-valued vector. Several techniques have been proposed to represent words in real-valued vectors, such as TF-IDF [10,45], Latent Semantics Analysis (LSA) [43], and neural network models such as word2vec [13] and GloVe [15].

Word vectorization is divided into global matrix factorizations such as LSA [46] and local context windows like the Skip-Gram model [13]. According to [10,43], both approaches have drawbacks. Global matrix factorization effectively utilized the statistical information, but it fails to capture word analogy. On the other hand, the Window-based model successfully captures word analogy but poorly uses global statistics.

The bag of words model is exploited by TF-IDF concepts [10,43]. TF-IDF relies on the statistical information of words from several documents. Terms refer to a word or a group of words; TF stands for term frequency. TF is basically the frequency of words in a document; to normalize the value, the frequency is divided by the number of words in a single document. IDF stands for inverse document frequency. The value of IDF is a logarithmic value of the number of documents divided by the TF.

GloVe [15] explores the global representation of the entire corpus and incorporates the meaning of the words in this. Word frequency and co-occurrence are the main metrics by which the values of real-valued vectors of particular words are calculated. GloVe is an unsupervised method, where there is no human to introduce ground-truth meaning into the collection of words (corpus). The basis of calculation is the exploitation of the frequency of particular words and the closest words surrounding each word.

In GloVe, the first step is collecting the most frequent words as the context. The second step is to scan the words in the corpus to build a co-occurrence matrix $X$. Let us consider $i$ as the index of frequent words and $j$ as the rest of the words in the corpus. $P_{ij}$ is the probability of a word $j$ occurring with the context word $i$.

$$P_{ij} = P(j|i) = \frac{X_{ij}}{X_i} \tag{1}$$

Let us consider two words $i$ and $j$, as well as a context word $k$; we can calculate a ratio of co-occurrence probability as follows:

$$F(w_i, w_j \widetilde{w}_k) = \frac{P_{ij}}{P_{jk}} \tag{2}$$

Finally, the loss function $J$ can be calculated as follows:

$$J = \sum_{i,j=1}^{V} f(X_{ij}) \left( W_i^T \widetilde{w}_j + b_i + b_j - log\, X_{ij} \right)^2, \tag{3}$$

where $f$ is the weighting function. The training aims to minimize the least squares error. Once GloVe is trained, every word is assigned to a particular real-valued vector.

### 3.3.2. fastText

Many strategies for incorporating morphological information into word representations have been proposed in recent years. Previous word embedding algorithms represented each word in the lexicon as a separate vector, with no shared parameters. In particular, they ignored internal word structure, which is a significant drawback for morphologically rich languages.

fastText, an open-source project by Facebook Research, is a fast and effective approach to learning word representation and performing text classification that is widely used for NLP. Instead of learning word representations, the fundamental goal of the fastText embeddings is to analyze the internal structure of words. This is especially effective in morphologically rich languages, as it allows learners to independently learn representations of distinct morphological forms of words [14].

Using Skip-Gram, the likelihood of the context, given by a word $t$, is parametrized by word vectors using a scoring function $s$:

$$s(w_t,\ w_c) = U_{w_t}^T V_{w_c} \tag{4}$$

With $u$ and $v$ chosen from the input and output embedding matrices. The scoring function in fastText is as follows:

$$s(w_t,\ w_c) = \sum\nolimits_{g\ \in\ G_{w_t}} Z_g^T V_{w_c} \tag{5}$$

$G_{w_t}$ represents the set of n-grams in word $w_t$, and $Z_g$ represents the vector of the $g$th n-gram. The vector of the context word $w_c$ is denoted by $V_{w_c}$.

### 3.3.3. Dataset

The data used for training and testing our model were adopted from the ASAD Dataset [19]. Massive Arabic datasets are needed to carry out pretraining using word embedding methods. These methods were adapted from [15] for GloVe word embedding and [47] for fastText word embedding. In this research, Glove was trained using the Arabic dataset ASAD. The ASAD dataset has 53,289 rows divided into Positive, Negative and Neutral sentiment polarities. The dataset is stored as a csv file. The dataset was dominated by the Neutral class (36,082 rows), followed by Negative and Positive, at 8674 and 8533 respectively. The dataset presents the issue of imbalance for model training. Imbalanced datasets will reduce the model's ability to recognize minority classes in classification tasks. Therefore, the minority class was subjected to random oversampling. In random oversampling, data from the minority class are reproduced at a random rate. In other words, data from the minority class are randomly reproduced based on the amount of data and the specified rate of oversampling. Several writers [45,46] concur that random oversampling might increase the chance of overfitting, since it duplicates the minority class instances exactly. As a result, the composition of the minority labels (Positive and Negative) was set at a similar number to that of the majority class (Neutral), at 36,082. Table 3 shows the initial composition before data augmentation and balanced composition due to random oversampling.

**Table 3.** Dataset composition before and after augmentation.

|                             | Neutral | Positive | Negative |
| --------------------------- | ------- | -------- | -------- |
| Initial composition         | 36,082  | 8533     | 8674     |
| Composition after augmentation | 36,082  | 36,082   | 36,082   |

The distribution of the dataset words is shown in Figure 2. The total number of words in the files is 524,334. The average number of words in the each tweet is 9.84, and the maximum word length is 26. The word distributions remain unchanged before and after the augmentation.

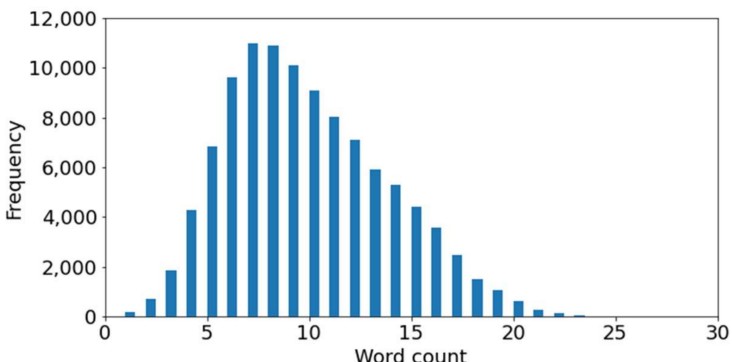

**Figure 2.** Dataset word distribution.

### 3.3.4. Long Short-Term Memory (LSTM)

Recurrent neural networks capture temporal relationships among words in a sentence. Textual information is time-series data where word order plays an essential role in defining the meaning of words and sentences. Long short-term memory is an implementation of recurrent neural networks with a special link between the nodes. Special components inside the LSTM unit include the input, output, and forget gates. Figure 3 shows a single LSTM unit.

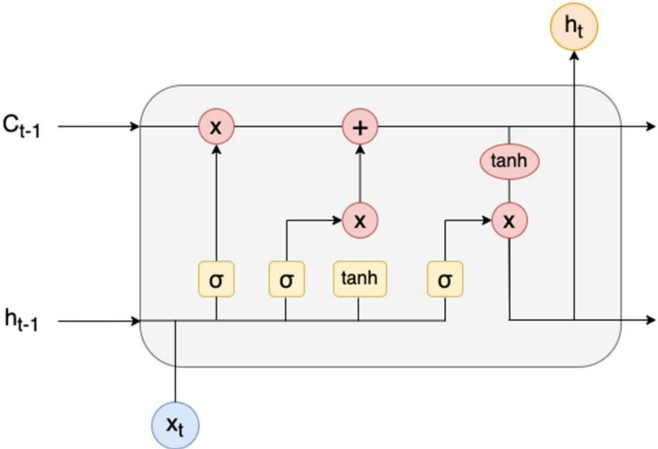

**Figure 3.** LSTM unit.

where

$X_t$ = Input vector at the time t.
$h_{t-1}$ = Previous Hidden state.
$C_{t-1}$ = Previous Memory state.
$h_t$ = Current Hidden state.
$C_t$ = Current Memory state.
[x] = Multiplication operation.
[+] = Addition operation.

In this research, we observe the behavior of single-layer, double-layer and triple-layer LSTM with 64 LSTM units and a final layer performing the classification into sentiment class using a dense layer with softmax function. The maximum input is 30 words, and this

is denoted as $X_t$, where t is equal to 1 . . . 30. Sentiment analysis is a problem with multiple inputs (n) and a single output. Figure 4 illustrates the LSTM architecture.

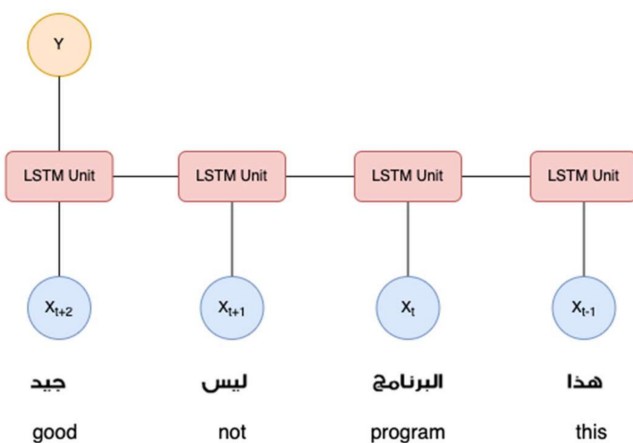

**Figure 4.** LSTM architecture.

Long Short-Term Memory is a module in an RNN network that solves missing gradient problems. In general, RNN applies the LSTM network in order to avoid propagation errors. This enables the RNN to learn over many time steps. LSTM contains cells that store information outside of a recurrent network. The cell is like the memory in a computer, deciding when the data needs to be stored, written, read, or deleted by means of the gate, as shown in Figure 4. There are four gates used by LSTM, including: input, forget, output, and new memory container.

This study uses four hyperparameters to effectively improve the performance of LSTM. The learning rate is 0.001. The Adam optimizer is utilized, since it has the property of adaptive learning levels [48]. Additionally, 64 LSTM units were used for training this model, although the number of units was dependent on the average length of each review. Finally, based on the previous embedding matrix process, a word vector size of 31,476,300 × 300 was used. In this research, we employed three architectures: single-, double- and triple-layer LSTM.

Figure 5 shows a single-layer LSTM in which 64 units are arranged in one layer. Several outputs are fed to a dropout layer, followed by a global max pooling layer, a dense layer, and three softmax layers on the top of the network. The expected output is classification as belonging to either the neutral, negative or positive class.

The details of the single-layer LSTM architecture are presented in Table 4 below.

**Table 4.** Single-layer LSTM architecture.

| Layer (Type) | Output Shape | Number of Parameters |
|---|---|---|
| Word Embedding | (None, 30, 300) | 31,476,300 |
| LSTM layer | (None, 30, 64) | 93,440 |
| Maxpooling | (None, 64) | 0 |
| Dense layer | (None, 64) | 4160 |
| Dense layer | (None, 3) | 195 |

Figure 6 shows the double-layer LSTM, where the input is an array of embedded words. Each input is accepted by each LSTM unit and followed by a dropout layer. The main difference between this and the single-layer architecture is the additional layer of the LSTM unit before the global max pooling and dense layers. As a result of the thicker architecture, there is an increased number of weighted parameters, leading to a heavier computing load.

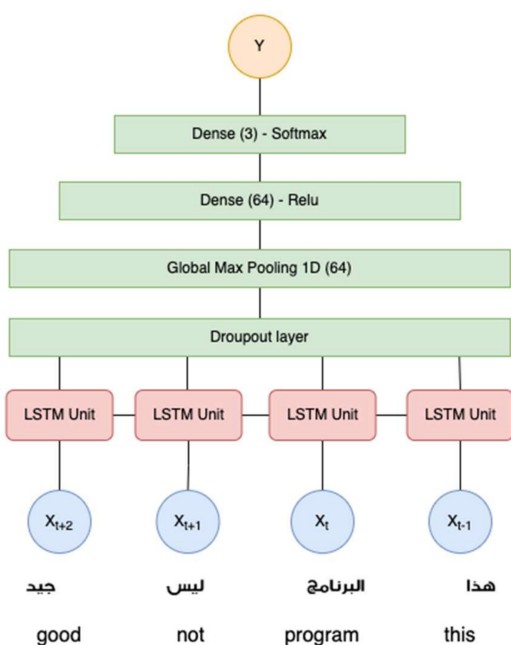

**Figure 5.** Single-layer LSTM architecture.

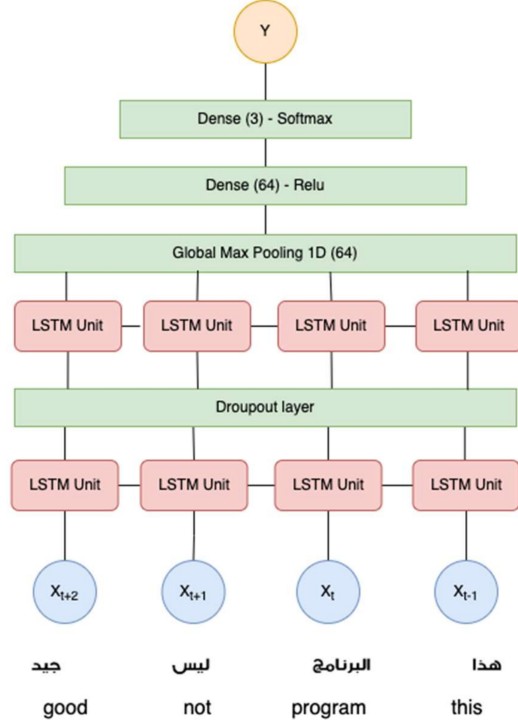

**Figure 6.** LSTM double-layer architecture.

Figure 7 shows the three layers of LSTM, where the input is an array of embedded words. The main difference between the single- and double-layer architectures is the new LSTM layer unit before the global max pooling and dense layers. This architecture is the thickest, with the most trainable weighting parameters.

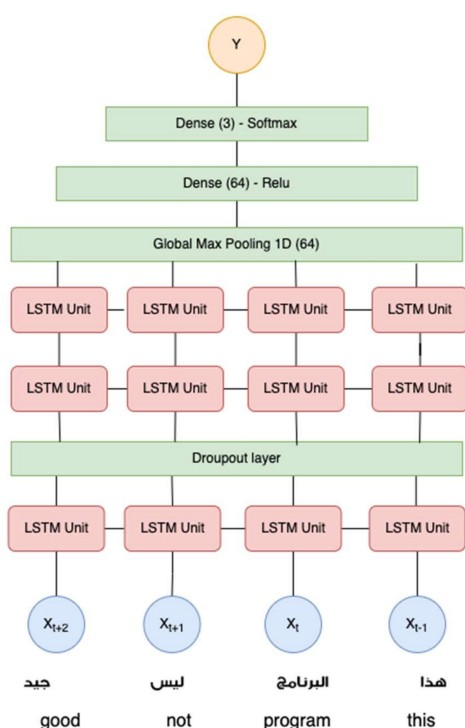

**Figure 7.** LSTM triple-layer architecture.

The details of the double-layer LSTM architecture are presented in Table 5.

**Table 5.** Double-layer LSTM architecture.

| Layer (Type) | Output Shape | Number of Parameters |
|---|---|---|
| Word Embedding | (None, 30, 300) | 31,476,300 |
| LSTM Layer | (None, 30, 64) | 93,440 |
| LSTM Layer | (None, 30, 64) | 33,024 |
| Max Pooling | (None, 64) | 0 |
| Dense Layer | (None, 64) | 4160 |
| Dense Layer | (None, 3) | 195 |

Triple-layer architectures are more complex, with more parameters needing to be trained, and they are expected to produce better accuracy. Table 6 presents the details of the triple-layer LSTM architectures.

**Table 6.** Triple-layer LSTM architecture.

| Layer (Type) | Output Shape | Number of Parameters |
|---|---|---|
| Word Embedding | (None, 30, 300) | 31,476,300 |
| LSTM Layer | (None, 30, 64) | 93,440 |
| LSTM Layer | (None, 30, 64) | 33,024 |
| LSTM Layer | (None, 30, 64) | 33,024 |
| MaxPooling | (None, 64) | 0 |
| Dense Layer | (None, 64) | 4160 |
| Dense Layer | (None, 3) | 195 |

## 4. Results and Discussion

This research aims to find the optimum word embedding method and the best LSTM architectures. Two embedding methods were evaluated for transforming the cleaned text into numerical vectors. LSTM is a recurrent neural network (RNN) variant with a more complex forget gate computation, allowing it to accommodate the influence of time-series signals from the immediate node and to retain the influence of a more extended sequence. In this research, we investigated three LSTM architectures: single layer, double layer, and triple layer, as shown in Figures 8 and 9. Training was carried out in 200 epochs to build the models.

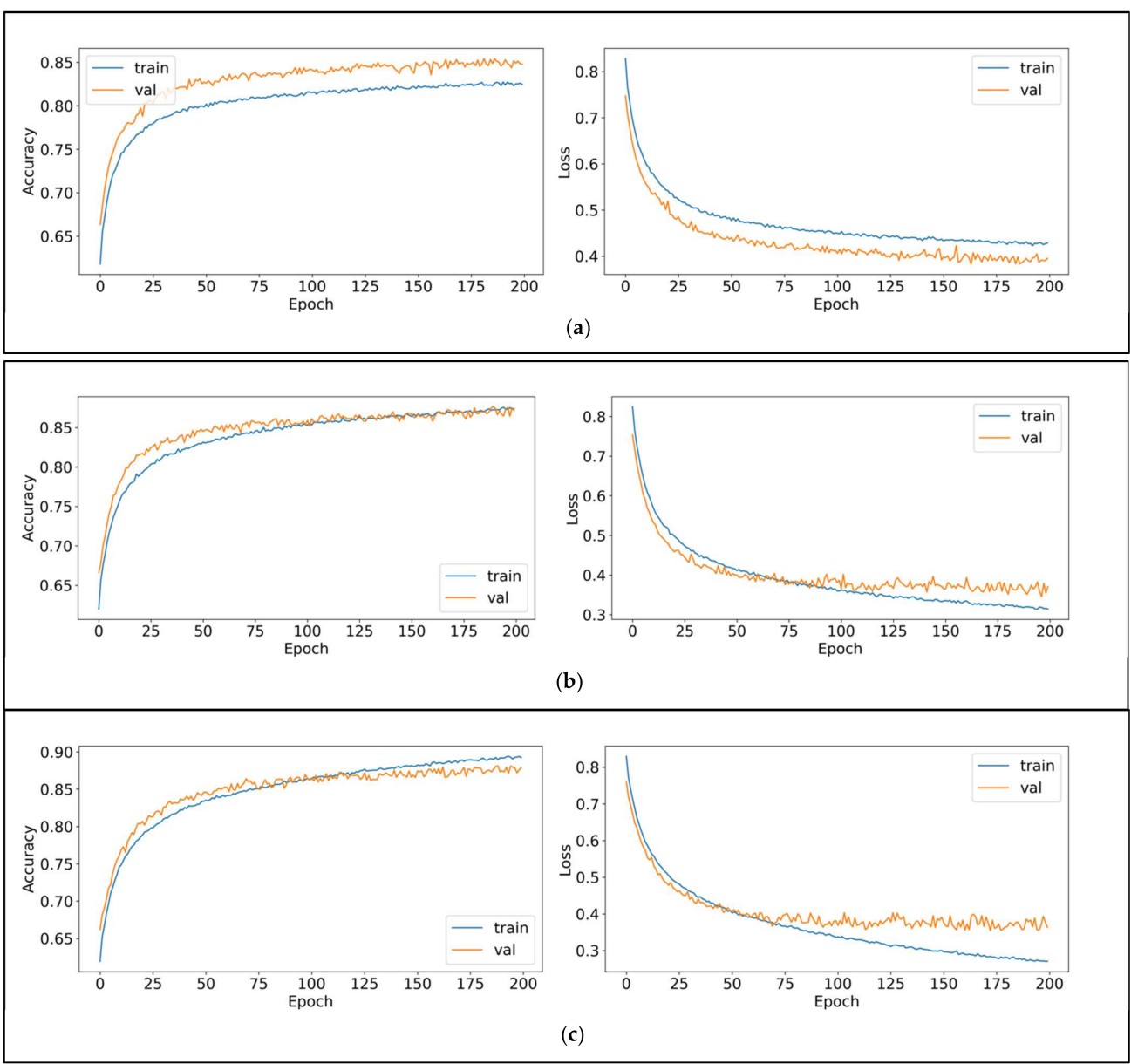

**Figure 8.** Training accuracy and training loss of the LSTM using GloVe embedding: (**a**) single-layer architecture; (**b**) double-layer architecture; (**c**) triple-layer architecture.

Figure 8 shows the single-, double- and triple-layer LSTM on the balanced dataset with GloVe embedding. Training accuracy and loss are plotted with the blue line, while yellow represents validation accuracy and loss. As can be seen, the single-layer LSTM shows a vast distance between the training and validation accuracy. In contrast, the double- and

triple-layer LSTM architectures show a smaller distance between the training and validation accuracy. Figure 8 demonstrates the effect of a thicker LSTM achieving convergency in earlier training iteration. Figure 9 shows the training progress on the next experiment scenario using fastText word embeeding.

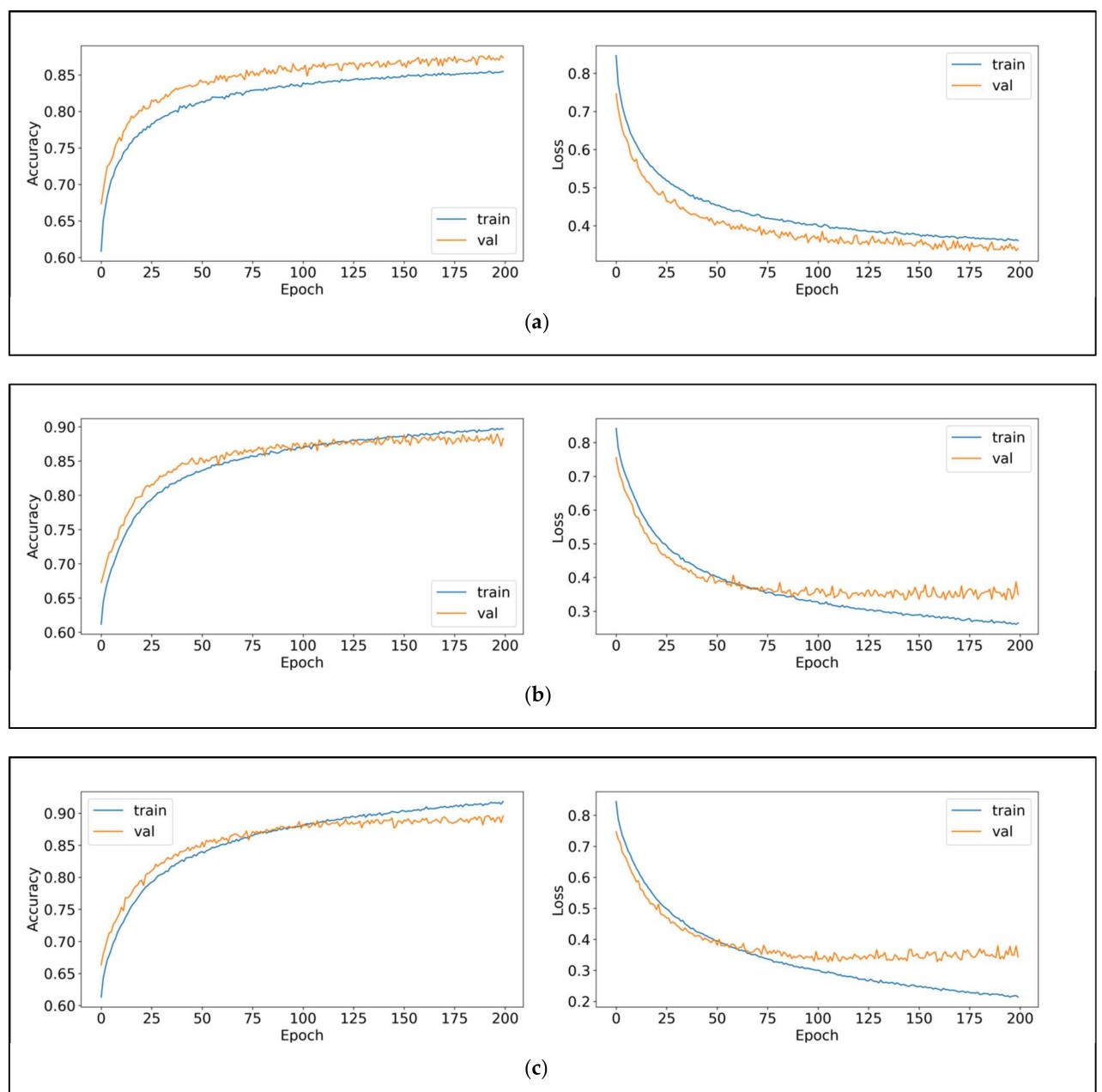

**Figure 9.** Training accuracy and training loss of the LSTM using fastText embedding: (**a**) single-layer architecture; (**b**) double-layer architecture; (**c**) triple-layer architecture.

Figure 9 shows the single-, double-, and triple-layer LSTM on the balanced dataset with fastText embedding. As can be seen, the single-layer LSTM results show a vast distance between training and validation accuracy. In contrast, the double- and triple-layer LSTM show a smaller distance between training and validation accuracy. As can be seen in double and triple layer of LSTM, the validation accuracy cross the training accuracy at earlier epoch of training. It is indicated that the model reached convergency earlier. This fact lead

us to implement early stopping technique in evaluation using the 5 fold cross validation. Table 7 presents the accuracy achieved by those architectures.

**Table 7.** Results of training, validation, and testing accuracy on 5 fold cross validation.

| | GloVe | | | fastText | | |
|---|---|---|---|---|---|---|
| | **Training Accuracy** | **Validation Accuracy** | **Testing Accuracy** | **Training Accuracy** | **Validation Accuracy** | **Testing Accuracy** |
| Single-Layer LSTM | 0.904 | 0.854 | $0.867 \pm 0.005$ | 0.928 | 0.876 | $0.881 \pm 0.006$ |
| Double-Layer LSTM | 0.943 | 0.876 | $0.891 \pm 0.008$ | 0.954 | 0.889 | $0.902 \pm 0.003$ |
| Triple-Layer LSTM | 0.955 | 0.881 | $0.903 \pm 0.008$ | 0.967 | 0.896 | $0.909 \pm 0.007$ |

In general, all of the accuracies achieved with fastText word embedding outperformed those achieved when using GloVe. Stacking more layers of LSTM led to higher accuracy in both training and testing. Triple-layer LSTM achieved better performance either double- or single-layer LSTM. This achievement, however, must be achieved at the expense of processing speed, as shown in Table 7. In this experiment, we used a CUDA GPU Titan V with a total memory of 12 GB. Table 8 compares the trainable parameters, training time, and testing time required for each LSTM design.

**Table 8.** LSTM architecture trainable parameters, and testing and training time.

| Architectures | Trainable Parameters | Training Time (min) | | Testing Time (µs) | | Testing in Consumer PC (s) | |
|---|---|---|---|---|---|---|---|
| | | **GloVe** | **fastText** | **GloVe** | **fastText** | **GloVe** | **fastText** |
| Single-Layer LSTM | 97,795 | 37 | 37 | 6.68 | 6.20 | 7.31 | 6.28 |
| Double-Layer LSTM | 130,819 | 79 | 78 | 6.91 | 6.90 | 10.8 | 11.9 |
| Triple-Layer LSTM | 163,843 | 117 | 117 | 7.39 | 6.91 | 16.6 | 16.4 |

Our results are in line with those of previous studies such as [28,41,42]. Although evaluations were performed on different datasets in previous works, the general trend shows the positive impact of LSTM. Regarding the embedding methods, fastText [14] demonstrates superiority compared to GloVe [15]. Our results confirm the findings reported by [41], where they demonstrated that Skip-Gram fastText performed better than Word2vec [13] and AraVec. In our experiment, fastText word embedding performed better than GloVe.

A greater number of trainable parameters leads to longer computation times, and the correlation among the trainable parameters and training time can be seen in Table 8. Training produces a model, which is compiled in the H5 file format. Testing uses the compiled model in a forward pass on the testing data. The elapsed time for each testing set consistently increases with the thickness of the implemented layers. Therefore, the triple-layer LSTM required a longer time than the other architectures. For the best achieved accuracy, with triple-layer LSTM using FastText word embedding, the training time of 1 h 25 min was recorded for the training dataset, with 6.91 µs testing time. In the implementation scenario, the input data consist of a single sentence, and therefore the required processing time is around 6.91 µs divided by 21,650. There will be no issue with the processing speed when implemented with server computer specifications.

We also tested the model using consumer personal computer (PC) with lower computer specifications in order to test its real implementation possibilities. The hardware specifications were: intel I5 with 8 GB memory. The testing time for 21,650 sentences was 16.4 s to achieve the best accuracy with fastText with triple-layer LSTM. Therefore,

every individual sentence required 0.75 milliseconds on average. According to those testing results and testing times, we consider that all of the observed architectures can be feasibly implemented.

## 5. Conclusions

This paper reports a comparison between two word-embedding methods in an Arabic sentiment analysis dataset (ASAD). In general, fastText provides a better vector for representing and recognizing sentiment in Arabic sentences. The accuracy achieved on sentences pre-processed with fastText was better than on those pre-processed with GloVe. Secondly, stacking the LSTM layers led to better accuracy, but, at the same time, decreased the speed of training and testing. In our experiment on the ASAD dataset, the best achievement was recorded using fastText with triple-layer LSTM, with accuracies of 96.7% and 90.9% for training and testing, respectively. Training and testing speed is significantly decreased when using the triple-layer LSTM architecture. The average testing time for a single sentence in a consumer PC is about 0.75 miliseconds. Sentiment analysis is still an interesting topic, since it varies between languages. This research provides an analysis of how the thicker LSTM layers improve the recognition rate and maintain relatively high implementation speeds, even in consumer PCs. Future research questions relate to the impact of thicker LSTM networks on accuracy and computational time. Experimenting with different bidirectional LSTM architectures is also interesting in order to see its impact on performance, since bi-LSTM has the ability to consider the impact of a word on previous words in a sentence as well as future words. With respect to the word embedding methods, comparing state-of-the-art embedding methods such as Word2vec and BERT will lead to a better understanding of acceptable word embedding techniques for the Arabic language. Observing the cross-language impact of word embedding methods and their impact on the performance of various classifiers is an interesting topic for determining feasible frameworks under various implementation conditions.

**Author Contributions:** Conceptualization, A.S. and F.A.; methodology A.S. and A.L.; software, M.H., A.L., I.O. and L.K.; investigation, A.S. and K.; writing—original draft preparation A.S., A.A. and N.A.; data curation, M.A. and F.A.; visualization, A.L.; validation, K.; supervision, K.; project administration, M.H.; funding acquisition, F.A.; resources, F.A. All authors have read and agreed to the published version of the manuscript.

**Funding:** This research was funded by the Deanship of Scientific Research at Imam Mohammad Ibn Saud Islamic University, grant number RG-21-51-01 and The APC was funded by Deanship of Scientific Research at Imam Mohammad Ibn Saud Islamic University.

**Institutional Review Board Statement:** Not applicable.

**Informed Consent Statement:** Not applicable.

**Data Availability Statement:** The authors used the publicly available dataset called Arabic Sentiment Analyis Dataset (ASAD). We provide the research results data and the code available for public.

**Acknowledgments:** The authors are grateful to LPPM Universitas Amikom Yogyakarta for their administrative support. We gratefully acknowledge the support of NVIDIA Corporation with the donation of the Titan V GPU used for this research.

**Conflicts of Interest:** The authors declare no conflict of interest.

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
