# Peer review of "Arabic Language Opinion Mining Based on Long Short-Term Memory (LSTM)"

_applsci, doi:10.3390/app12094140_

Round 1

Reviewer 1 Report

Congratulate authors for their work.

The paper reports good study on Arabic translation. As future scope of study, authors could replicate the model and measure performance on higher configuration computation node. 

Author Response

Dear Respected Reviewer,

I would like to express my gratitude for your valuable review. We would like to revise your review comments accordingly. 

Regards 

Arief Setyanto

Reviewer 2 Report

The paper is well written. But I have some suggestion before publishing it:

1. Check and correct spelling in the following lines:

Line 22, “united nation” must be “United Nations”

Line 35, “Terms Frequency – _Invers Text Frequency (TF-IDF)” must be “Term Frequency – Inverse Document Frequency (TF-IDF)”

Line 60, ”… the supervised approach needs many labelled corpora to train the classifier.” must be “… the supervised approach needs manually labelled corpora to train the classifier.”

Line 70, “Deep learning has made a successful breakthrough on many recognitions tasks such as …” must be “Deep learning has made a successful breakthrough on many recognition tasks such as …”

Line 260, “TF basically the frequency of words in a document, …” must be “TF is basically the frequency of words in a document, …”

Line 357, “…information outside a recurring network.” must be “…information outside a recurrent network.”

Line 477, “The average of testing time for a single sentence in consumer PC ia …” must be “The average of testing time for a single sentence in consumer PC is …”

Line 481, “Future research question about the impact of the thciker LSTM network …” must be “Future research question about the impact of the thicker LSTM network …”

Line 483, “LSTM is also intereting to see the impact to the performance since bi-LSTM has an ability …” must be “LSTM is also interesting to see the impact to the performance since bi-LSTM has an ability …”

Lines 486-487, “… to the performance of various classifier is an interesting topics t find out the feasible framework in various implementation condition.” must be “… to the performance of various classifier is an interesting topics to find out the feasible framework in various implementation condition.”

2. Check the dimensions of figures 8 and 9 in accord to the textwidth.

3. Author contributions should be written based on the manuscript template.

4. In References section please refer to the MDPI Reference List and Citations Style Guide.

Author Response

Dear Reviewer,

We appreciated very much the encouraging, critical and constructive comments on this manuscript by the reviewer. The comments have been very thorough and useful to improve the manuscript. We strongly believe that the comments and suggestions have increased the scientific value of revised manuscript. We are submitting the revised manuscript in response to all the reviewer’s comments accordingly. 

Regards 

Arief Setyanto

Reviewer 3 Report

Dear authors, this is my comments

  1.  first, the authors should write the valid reasons why you investigated the effect of word embedding algorithms by only choosing Glove and fastText because there are many other new word embedding algorithms out there. (refer to 3.1 frameworks in page 6)
  2.  in section 3.4, for the fastText, it is implicitly clear that it was trained from Arabic language (I have checked the title of [51])  but GloVe is not clear that it was trained from Arabic language or not. Do you use pretrained Glove or self-trained Glove from Arabic language? and which dataset you have used to train word embedding? elaborate more details on training word embeddings will be clearer.
  3. in section 3.5 the authors should state the reasons why we need to stack LSTM model and why not stacked Bi-LSTM? is it possible to investigate stacked Bi-LSTM in the future or not? (This could be add in the future work)
  4. for the result, I think it must use early stopping technique to find the exact number of epoch as seen in Figure 8.b Loss graph at epoch 80 it showed overfitting and in Figure 8.a it seems it can continue to train more epoch until the valid graph cannot improve. Each model has different the certain epoch.  After stopping training with the right epoch, then final report the accuracy with the "TEST SET. Each model should run at least 30 or more experiments to find the average accuracy and report with standard errors. But for graph plotting purpose, 200 epoch it seems fine for me to show what's going on when training across different models. But when we want to report the final test accuracy it must use proper epoch number, "early stopping" technique may help in this situation. 

Author Response

(The authors gave the same response as above.)

Round 2

Reviewer 3 Report

Authors have fixed and explained the existing issues with clear details. Congrats to the authors.